# Chondrosarcoma: New Molecular Insights, Challenges in Near-Patient Preclinical Modeling, and Therapeutic Approaches

**DOI:** 10.3390/ijms26041542

**Published:** 2025-02-12

**Authors:** Lorena Landuzzi, Francesca Ruzzi, Pier-Luigi Lollini, Katia Scotlandi

**Affiliations:** 1Experimental Oncology Laboratory, IRCCS Istituto Ortopedico Rizzoli, 40136 Bologna, Italy; katia.scotlandi@ior.it; 2Laboratory of Immunology and Biology of Metastasis, Department of Medical and Surgical Sciences (DIMEC), University of Bologna, 40138 Bologna, Italy; francesca.ruzzi2@unibo.it; 3IRCCS Azienda Ospedaliera Universitaria di Bologna, 40138 Bologna, Italy

**Keywords:** chondrosarcoma, preclinical models, targeted therapies, immunotherapy, multi-omic profiling

## Abstract

Chondrosarcoma (CS), the second most common malignant bone tumor after osteosarcoma, accounts for 20–30% of all malignant bone tumors. It mainly affects adults, middle-aged, and elderly people. The CS family includes various entities displaying peculiar biological, genetic, and epigenetic characteristics and clinical behaviors. Conventional CS is the most common subtype. High-grade, dedifferentiated, and mesenchymal CS, as well as unresectable and metastatic CS, exhibit poor prognoses due to their intrinsic resistance to radiotherapy and chemotherapy, underscoring the urgent need for novel therapeutic strategies. CS research is dealing with several challenges. Experimental studies can rely on animal and patient-derived models, but the paucity of representative near-patient preclinical models has hampered predictive drug screening research. This review describes the main clinical and molecular features of CS subtypes, discussing recent data on the genetic alterations and molecular mechanisms involved in CS pathogenesis and progression. The review provides an overview of the current in vitro and in vivo CS models, discusses their advantages and limitations, and highlights the recent efforts in the development of new targeted therapies against CS dependencies, including IDH1/2 mutations, NAD^+^ dependency, and SIRT1-HIF-2α axis, or exploring DR5 targeting, antiangiogenic therapies, epigenetic drugs, and immunological approaches. All such strategies, in combination with advanced preclinical modeling and personalized multi-omic profiling, hold promise for improving the survival of patients with advanced CS.

## 1. Introduction

Chondrosarcoma (CS) accounts for 20–30% of all malignant bone tumors and ranks second among the primary malignant tumors of bone after osteosarcoma (OS). In contrast to OS, which affects children and adolescents, CS mainly arises in adults, middle-aged (>40 years old), and elders [1,2]. CS is a collective general definition for a family of different entities showing different biology, genetics, and epigenetics. The incidence of CS, and specifically of atypical cartilaginous tumor (ACT), the low-grade variant, has increased in recent years. This could be due to both an aging population, since CS incidence increases with age, and intensified diagnostic imaging activity [3]. Given the growing proportion of elderly people in industrialized countries, it is expected that clinical management of CS will hereafter represent an increasing challenge. Most CS patients show a good outcome after adequate wide surgical treatment because CS is a slow-growing tumor that rarely metastasizes. In contrast, advanced, metastatic, or unresectable CS patients have a dismal prognosis due to innate resistance to chemotherapy and radiotherapy and to the very few therapeutic opportunities that are currently available for these patients. This review will summarize the clinical and molecular features of the different CS subtypes, mainly focusing on conventional CS. We will highlight the recent findings on the genetic and molecular alterations related to CS progression, high grade, and dedifferentiated CS, addressing, among others, the role of IDH1/2 mutations, the EPAS1 gene (encoding HIF-2α) amplification, and SIRT1-HIF-2α axis activation. We will outline the advances in the development of representative near-patient CS preclinical models and their current limitations. Moreover, we will provide an overview of the different potential innovative therapies, targeting CS dependencies or exploiting immunological approaches, that hopefully might help to overcome treatment failures in CS.

## 2. CS Histological Classification and Clinical Aspects

In the intricate landscape of the CS family tumors, many different subtypes with peculiar biological and clinical behavior can be recognized. Conventional CS is the most frequent histotype, accounting for 85–90% of the cases. CS is classified as primary when it arises without any pre-existing lesions, and as secondary when it develops from a benign cartilaginous tumor, such as enchondroma or osteochondroma. Depending on its location within the bone, CS can be categorized as central or peripheral. Central CS arises within the medullary cavity, while peripheral CS develops from the surface of the bone, such as from the cartilage cap of an exostosis. There is also a rarer type called periosteal CS, which originates from the periosteum. Primary CS is typically central, whereas secondary CS can be either central or peripheral [4,5]. Peripheral CS arises, in nearly 50% of the cases, in patients carrying a heritable predisposition to form multiple osteochondromas or exostoses, which are caused by the presence of a dysfunctional germline allele of exostosin (EXT) 1 or 2, and it represents one of the few sarcoma types that arise from a detectable precursor lesion in the context of heritable syndromes [6].

Conventional CS, in contrast to OS, which is mostly diagnosed as a high-grade malignancy, is usually diagnosed as a localized disease with grade 1 or 2, and it is commonly a slow-growing tumor characterized by hyaline cartilaginous differentiation. Conventional CS occurs mainly in the pelvic bone, femur, proximal humerus, and scapula. Recently, grade 1 CS of long and short tubular bones, but not those of flat bones, pelvis, scapula, and skull base, has been renamed as atypical cartilaginous tumor (ACT) underlining its locally aggressive behavior but low malignancy [7]. The conventional CS grade 3, which can be highly metastatic, is much less frequent (5–10% of the cases). Grade 4 corresponds to the dedifferentiated CS and accounts for 10% of the cases. Dedifferentiated CS shows components of low-grade CS and abrupt transition to areas of high-grade sarcoma, most frequently osteosarcoma, but also fibrosarcoma or undifferentiated pleomorphic sarcoma.

Other, rarer entities are mesenchymal CS (2%), affecting mainly adolescents and young adults and exhibiting highly malignant behavior [8], and clear cell CS (2–6%), which is usually considered a low-grade variant but can present late metastases [9,10]. The extraskeletal myxoid CS, which accounts for 3% of all soft tissue sarcomas [11,12] and represents an intermediate-grade tumor previously classified as a cartilaginous tumor, is now included among soft tissue sarcomas, and it is defined as a tumor of uncertain differentiation.

Dedifferentiated CS and mesenchymal CS show the worst prognosis having a more aggressive growth and a higher risk of metastases. In addition to histopathological features (for a review see the article of Zając et al. [13]), differential molecular pathological diagnosis for CS subtypes can take advantage of the immunohistochemical or genetic detection of isocitrate dehydrogenase (IDH)1/2 mutations found in 50–80% of central, or periosteal CS and dedifferentiated CS, or exostosin (EXT)1/2 mutations found in peripheral CS. Conversely, mesenchymal and clear cell CSs are negative for IDH1/2 mutations and show positivity for S100, SOX9, and Bcl2 [7]. In mesenchymal CS and extraskeletal myxoid CS, specific chromosomal translocation can be detected [13,14]. CS subtypes and their main molecular and clinical features are summarized in Table 1.

The mainstay of CS clinical management remains the wide-margin surgical resection. ACT is a low-grade lesion that in cases of an intraosseous lesion can be surgically treated with extended curettage and local adjuvants [3]. The biological features of CS, characterized by low proliferation rate, overexpression of P-glycoprotein mediating multi-drug resistance, poor vascularization, and hyaline cartilaginous matrix, make CS intrinsically resistant to both chemotherapy and radiotherapy. In the tumor microenvironment of CS, more than in other tumors, a prominent role in tumor growth, progression, and drug sensitivity is played by the extracellular matrix, which is a dense cartilaginous matrix, mainly composed of collagen type II fibers and hyaluronic acid, that strongly affects the crosstalk between dispersed clusters of tumor cells and stromal, endothelial, and immune cells, and interferes with the diffusion of cytokines, chemokines, angiogenic factors, and antitumor drugs.

For advanced conventional CS, chemotherapy usually entails treatment with doxorubicin alone or in combination with cisplatin or ifosfamide, but median progression-free survival (PFS) in patients with an advanced disease does not reach 4 months and survival rates did not substantially improve over the last decades [7,14,15].

For dedifferentiated CS, combining intensive chemotherapy with surgery yielded positive survival results compared with previous data, and current guidelines indicate the use of OS chemotherapeutic protocols when appropriate, based on the age of the patient. In the case of mesenchymal CS, adjuvant and neoadjuvant chemotherapy regimens, similar to those used for Ewing sarcoma, have shown an increase in survival rate and are currently recommended by the National Comprehensive Cancer Network (NCCN) and by the European Society of Medical Oncology (ESMO) [5,7,16]. The most common predictors of local and distant relapse and survival are histological grading and adequate surgical margins. For conventional CS the 5-year survival rates are 90%, 81%, and 29%, for grades 1, 2, and 3, respectively. Local recurrences are frequent, being estimated at around 50% for low-grade tumors and around 80% for high-grade tumors. The main efforts of surgical resection are aimed at adequate negative surgical margins. Surgical margins are related to local recurrence in all CS grades. Wide margins can be better achieved in CSs of the extremities than in pelvic tumors. Indeed, the risk of local recurrence increases when the tumor site hampers radical resection. Overall pelvic location, narrow surgical margins, and a high grade are directly correlated with an increased risk of local recurrence [17]. Typically, progression to a higher grade is observed upon local or distant recurrence, and the lungs are the most common site of metastasis [18]. Other, less frequent, metastatic sites include the liver, lymph nodes, brain, and soft tissues. Patients that display indolent tumors and complete local control of the primary sarcoma with no signs of systemic disease other than isolated lung metastases can be considered for pulmonary metastasis resection or radio-frequency ablation and potential chemotherapy, based on the patient’s overall clinical performance status [19]. Even repeated pulmonary metastasectomy can improve survival despite recurrent disease [20]. However, disease control, due to resistance to conventional anticancer treatments, is difficult to achieve in case of multiple extra-pulmonary metastatic lesions or inoperable locations.

## 3. CS Genetics

CSs present complex karyotypes and genetic instability. Genetic data on CSs include several anomalies with alteration in chromosome numbers and ploidy status, gain and loss of chromosomes, and chromosomal rearrangements. Central and peripheral CSs differ at the genetic level with either isocitrate dehydrogenase (IDH)1/2 mutations or exostosin (EXT)1/2 mutations, respectively.

Germline mutation and functional loss of the tumor suppressor genes EXT1 or EXT2, catalyzing heparan sulfate polymerization and participating in the differentiation of chondrocytes, are commonly found in multiple osteochondromas and predispose to the development of secondary peripheral CS. In addition, sporadic secondary CSs frequently display loss of heterozygosity at the EXT1 and EXT2 loci, while primary CSs usually do not [21].

High-grade and dedifferentiated CSs harbor additional oncogenic alterations common to many types of sarcomas, such as inactivation of CDKN2A/2B, TP53, and RB1, alterations of hedgehog pathways, alterations in genes related to telomere regulation such as TERT gene amplifications, TERT promoter mutations, and alpha thalassemia/mental retardation syndrome X-linked (ATRX) mutations [22,23]. The mutation of COL2A1, encoding the collagen type II alpha 1 chain of type II collagen fibers, the major constituent of articular cartilage, can be frequent. COL2A1 mutations have been identified in 37% of CSs [24] but they were not predictive of prognosis.

Some tumor histotypes display distinctive chromosomal translocations and mutations. For example, mesenchymal CS exhibits recurrent translocations such as HEY1::NCOA2 or IRF2BP2::CDX1 driving tumorigenesis [8,25].

Genetically distinct from osseous CS is the extraskeletal myxoid CS, which is characterized by the chromosomal translocation t(9;22)(q22;q12), or more rarely t(9;17)(q22;q11) and t(9;15)(q22;q21), involving the nuclear receptor subfamily 4, group A (NR4A3) and giving rise to the fusion genes EWS::NR4A3, TAF15::NR4A3, and TCF12::NR4A3, respectively [11].

### 3.1. IDH1/IDH2 Mutations

The detection of isocitrate dehydrogenase (IDH1 and IDH2) heterozygous mutations, affecting only single alleles of the gene, in 50–70% of enchondromas and conventional central and periosteal CS but not in other mesenchymal tumors, fostered the understanding of CS natural history and malignancy [26]. IDH1/2 mutations can be detected also in 50–80% of dedifferentiated CS, giving rise to a peculiar methylation profile distinguishing conventional CS from dedifferentiated CS [27]. Conventional secondary peripheral CSs and osteochondromas, characterized by mutations in EXT1 and EXT2, do not harbor IDH1/IDH2 mutations. IDH1 and IDH2 encode for isocitrate dehydrogenases, enzymes that are located in cytosol and mitochondria, respectively, and which normally catalyze the oxidative decarboxylation of isocitrate to generate α-ketoglutarate (αKG). IDH1 mutations usually result in substitutions at R132, with R132C as the predominant variant (around 60%) in CS, whereas IDH2 mutations affect either R172, analogous to R132 in IDH1, or R140 [28,29,30]. Mutant IDH1 and IDH2 enzymes fail to convert isocitrate to α-ketoglutarate and gain a new function that leads to the conversion of αKG to D-2-hydroxyglutarate (D-2HG). The IDH1 R132C variant is one of the most efficient D-2HG producers [30]. D-2HG is an oncometabolite that competitively inhibits αKG dependent enzymes, including histone demethylases, tet methylcytosine dioxygenase 2 (TET2), and hypoxia-inducible factor (HIF) prolyl hydroxylases mediating histone hypermethylation, DNA hypermethylation, and HIF-1α overexpression. Dysregulation of chromatin methylation and widespread epigenetic modifications foster the initiation and progression of CS. Histone and DNA hypermethylation are assumed to prevent the expression of genes related to full chondrocytic differentiation. Interestingly, the introduction of an IDH2 mutant transgene in mice was able to induce enchondroma-like lesions [31]. IDH1/IDH2 mutations can act by increasing the levels of HIF-1α and the associated transcriptional activity of its target genes, related to the adaptation of cells to low oxygen tension and involved in angiogenesis, invasion, and glucose metabolism, functions that are relevant to tumor growth, progression, and chemotherapy resistance. High levels of HIF-1α and glycolysis-associated genes have been reported in high-grade CS compared to low-grade CS.

The relevance of IDH mutation concerning CS patients’ overall survival is controversial. IDH mutations were associated with worse outcomes in some cohorts of patients [32,33], but opposite results were reported by other studies [15,34]. A recent work reported an association of IDH mutations with longer relapse and metastatic-free survival in high-grade CS, but no significant correlation with overall survival [23]. The relevance of IDH mutation as a reliable predictor of prognosis in CS needs to be further investigated in larger and multicentric cohorts of patients.

IDH1/2 mutations are druggable targets and are already exploited for therapeutic purposes in gliomas, cholangiocarcinoma, and acute myeloid leukemia (AML) [28,35], but, as we will see later, their significance as targets in CS clinical management is still under evaluation.

### 3.2. TP53 Mutations

Mutations in TP53 are the second most prevalent alteration (around 20–40%) of CS after IDH1/2 mutations. Several studies reported that TP53 mutations were more frequent in high-grade and dedifferentiated CS. Denu et al. found, in their cohort of CS patients, that TP53 mutations were associated with worse overall survival and metastasis-free survival, and with a higher risk of local recurrence and metastatic relapse [15]. Several data suggest that TP53 status can help in stratifying CS patients at high risk of relapses and progression.

### 3.3. EPAS1 Gene Amplification and Hypoxia-Inducible Factor-2α (HIF-2α) Upregulation

HIF-2α (encoded by EPAS1 gene) is a transcription factor that, like HIF-1α (encoded by HIF1A gene), modulates the hypoxic response. However, the two hypoxia-inducible factors play distinct roles in hypoxic gene regulation and tumor progression. In a study by Kim and colleagues [36], HIF1A gene amplification did not show a correlation with overall survival rates or disease-free survival rates in CS patients. In contrast, the amplification of the EPAS1 gene was significantly associated with decreased overall survival rates and increased dedifferentiation. High-grade CS exhibits elevated levels of HIF-2α, but the underlying mechanisms driving this upregulation remain unclear. Although IDH mutations partially contribute to the increased stability of HIF-2α in CS cells, a comprehensive understanding of the mechanisms is still lacking. Notably, HIF-2α has been implicated in drug resistance and identified as a downstream effector of Sirtuin1 (SIRT1) in CS [36]. SIRT1, a nicotinamide adenine dinucleotide (NAD)^+^-dependent protein deacetylase, is predominantly located in the cell nucleus. It deacetylates both histone and non-histone proteins, influencing the activity of key regulatory pathways, including the nuclear factor-κB family, p53 family members, and FOXO transcription factors [37]. Through these transcription factors, SIRT1 is involved in controlling energy metabolism, epithelial-mesenchymal transition, and metastasis. SIRT1 expression is significantly increased in high-grade and dedifferentiated CS compared to low-grade tumors and is correlated with poor prognosis in CS patients. Suh et al. [38] found that the silencing of SIRT1 induced inhibition of tumor growth in vivo, both in an orthotopic model of the SW1353 and in a subcutaneous model of the JJ012 human CS cell lines. In vitro, glucose deprivation caused activation of SIRT1 and stabilization of HIF-2α in SW1353 and JJ012 cells, increasing tumor cell survival compared to SIRT1-silenced cells. These data suggested that CS cells can survive the low-nutrient supply condition in the cartilaginous tumor microenvironment and the NAD^+^-dependency by activating the NAD^+^-SIRT1-HIF-2α axis able to upregulate NAD^+^ biosynthesis. This was consistent with the finding that high-grade CS tumors are characterized by the up-regulated expression of NAD^+^ biosynthesis genes such as nicotinate phosphoribosyltransferase (NAPRT), nicotinamide phosphoribosyltransferase (NAMPT), and indoleamine2,3-dioxygenases (IDO). When CS cells were genetically modified to overexpress NAPRT or NAMPT, their resistance to doxorubicin increased, highlighting the role played by NAD^+^ upregulation in the chemoresistant behavior of CS cells. The use of inhibitors of NAD^+^ biosynthesis enzymes and most of all SIRT1 was crucial for CS cell survival, inducing a synergistic cytotoxic effect when combined with chemotherapy [38], therefore providing a rationale for the development of new therapeutic approaches targeting the SIRT1-HIF-2α axis to enhance CS chemosensitivity.

### 3.4. MYC Amplification in CS

MYC amplification is frequently found in highly aggressive tumors. MYC amplification has been detected in 15%, 20%, and 21% of grade 2, 3, and dedifferentiated CS, respectively, while no amplification was observed in enchondromas and grade 1 CS. In the human genome, the MYC gene is located on chromosome 8. In CS, it was noted that the frequency of polysomy 8 increased with the tumor grade, being 18% in grade 1, 31% in grade 2, 80% in grade 3, and 29% in dedifferentiated CS.

MYC amplification or polysomy 8 correlates significantly with poor outcomes and short survival of CS patients [39]. Different molecules for direct and indirect MYC therapeutic targeting are currently under investigation [40] and in the future may have therapeutic relevance at least for the subgroup of MYC overexpressing CS. Of note, a dual inhibitor of bromodomain-containing protein 4 (BRD4) and PI3K-Akt-mTOR, SF2523 blocked Akt-mTOR activation and downregulated MYC and Bcl2 expression in the CS cell line SW1353 and in vivo strongly impaired its subcutaneous tumor growth in immunodeficient mice [41].

### 3.5. CS Multi-Omic Profiling

Recently a CS multi-omic signature, deriving from mRNA expression, microRNA, and DNA methylation profiling systems has been proposed to detect high-risk CS patients. Based on one of the largest genetically characterized cohort of human CS reported to date, the multi-omic classification highlighted the combined effect of the acquisition of high levels of cell cycle-related genes, the silencing of the 14q32 imprinted locus related to the downregulation of several microRNAs (including miR-154, miR-382, and miR-384, previously shown to inhibit tumor growth in bone sarcomas) and the hypermethylation of DNA at a genome-wide level, induced by IDH mutations, in the acquisition of a more aggressive grade and worse prognosis [42]. The combined signature identified three groups with favorable prognosis (IDHwt/14q32high, IDHmut/14q32high, and IDHwt/14q32low), two groups with an intermediate prognosis (IDHmut/14q32low and Prolif high) and the dedifferentiated group (14q32low/Prolif high) with the worst prognosis. The integrated molecular evaluation of CS seems to be more reliable than the simple determination of IDH status in prognosis prediction and probably can partially explain previous controversial results.

By single-cell RNA sequencing, Su et al. identified four CS signatures based on the expression of proliferation, stromal, or leukocyte-related genes. High-grade and dedifferentiated tumors were characterized by a high proliferation index, and the immunosuppression index further distinguished the dedifferentiated group from the high-grade group. Active immune response index identified low-growing tumors [43]. In the same study, endoplasmic reticulum (ER) stress regulators, such as DNA Damage Inducible Transcript 3 (DDIT3, also named CHOP) and HSPA5 emerged as markers for overall survival in conventional central CS patients [43]. Upregulation of DDIT3/CHOP in malignant tumors compared to benign lesions was confirmed by immunohistochemistry. High expression of DDIT3/CHOP or HSPA5 was correlated with poor prognosis. In addition, in a CS patient-derived xenograft mouse model, induction of ER stress boosted tumor growth, while inhibition of ER stress suppressed tumor progression, suggesting the targeting of ER stress as a new potential therapeutic strategy in CS.

## 4. CS Preclinical Models and Near-Patient Models

### 4.1. CS Cell Lines

Establishing CS cell lines is challenging due to their typically slow growth. Additionally, there are few permanently growing cartilage cell lines that exhibit a cartilage phenotype [44,45,46]. The Cellosaurus–Cell Line Encyclopedia (accessed on 18 September 2024) displays 82 entries for truly human CS cell lines and 10 entries for mouse, rat, and hamster models. Of note, no tumorigenic mouse CS cell lines are currently available for experimental studies [47]. A canine CS cell line endowed with tumorigenic ability in nude mice was recently reported [48]. Some human CS cell lines such as JJ012, SW1353, and ch-2879 [49,50] have been extensively used in the study of CS molecular biology, mechanisms of tumorigenesis, and drug sensitivity [51]. While the currently available panel of CS cell lines represents several histological subtypes, it falls short of fully capturing the clinical heterogeneity of CS. This limitation is particularly critical for a drug-resistant tumor like CS, making effective tumor modeling an ongoing and significant challenge. The most used and characterized CS cell lines and their biological and molecular features are listed in Table 2.

### 4.2. Organoids and 3D Models for CS

Given the difficulties in establishing continuous and functional CS cell lines, the search for relevant in vitro models that replicate the complexity of the tumor microenvironment and better predict in vivo biological behavior led to extensive exploration of 3D models and organoids for CS. Methods for creating 3D cultures from CS cell lines include scaffold-free techniques, such as hanging drop and low-attachment systems, scaffold-based approaches using materials such as alginate, collagen, and titanium beads, and levitational assembly in high magnetic fields [63,64]. In all these studies, 3D CS spheroids showed higher chemoresistance and radiation resistance as compared to 2D cultures. Previous studies in 2D culture conditions showed that talazoparib, a polyADP-ribose polymerase (PARP) inhibitor, sensitized CS cell lines to chemotherapy and radiotherapy. When using an alginate 3D spheroid model, the synergic activity in reducing CS spheroid growth was evident only after long-term drug exposure. The combination with radiotherapy was less effective, showing inhibition only in one out of three CS cell lines tested. The 3D spheroid model, compared to the 2D model, offers more stringent conditions closely resembling CS-resistant behavior in vivo [65] and can therefore provide more predictive results. Resistance was related to higher expression in the 3D setting of genes involved in cartilaginous matrix production, and activation of the multi-drug resistance pump. Moreover, a different study, using non-contact co-culture assays with CS 3D spheroids and monocytes, found that CS spheroids elicited monocyte polarization into pro-tumoral M2 macrophages and reciprocally, macrophages induced a size increase in CS spheroids indicating a pro-tumoral crosstalk [66].

In a recent study, patient-derived tumor organoids (PDTO), established from 21 different types of sarcomas including CS, were obtained. For most sarcoma types, PDTO responses to treatment correlated with patient therapy outcomes and provided actionable drug sensitivity data, supporting personalized medicine strategies. Although only a small number of CS samples were included, they accounted for the majority of the highly chemoresistant cases in the study, consistent with the well-known drug-resistant phenotype of CS [67].

### 4.3. CS Chorion-Allantoic Membrane (CAM) Models

CAM models use chicken eggs from day 4 after fertilization to days 15–16. Cell seeding usually takes place on days 7–10 and is observed up to days 15–16. Chicken embryos become immunocompetent on day 18. Their natural immunodeficiency in the first two weeks allows transplantation from different tissues and species [68]. Metastatic growth can be assessed by intravenous injection of the cells in the CAM veins of 11-day-old chick embryos. Patient-derived musculoskeletal sarcomas, including CS, can be engrafted on the CAM to produce small patient-derived tumors suitable for angiogenesis studies, functional studies, and short-term testing of therapeutic agents. Guder and colleagues reported that macroscopic patient-derived CS tumor xenografts did not increase in size regardless of tumor entity, while primary cell culture-derived xenografts, including CS cell cultures, showed tumor growth in CAM [69,70].

Of note, embryonal models are not considered animals, nor are they regulated by laws on experimental animal protection, therefore they do not require institutional authorizations and bureaucratic burden [71]. CAM models offer a rapid and cost-effective tool; however, the tumor growth is limited in dimensions and observation time. Some concerns are related to their use as a transplantation model for mammalian tumors since they do not completely reproduce the mammalian host and its tumor microenvironment and metabolism.

### 4.4. CS In Vivo Animal Models

In vivo modeling of chondrosarcoma aims to deepen our understanding of its development and progression while providing robust platforms for preclinical drug testing in this rare and heterogeneous disease. Animal models of CS include older models based on spontaneous tumors and more recent ones utilizing transgenic mice engineered to develop CS. Human-derived CS models have also been established through the use of immunodeficient mice. While animal and human-derived CS models exhibit distinct characteristics, each with specific advantages and limitations, both have significantly contributed to advancing our knowledge of CS pathogenesis and facilitating the investigation of novel therapeutic approaches.

### 4.5. Spontaneous Transplantable CS Animal Models

Among the animal transplantable CS models, one of the most studied was the Swarm rat chondrosarcoma (SRC). It was derived from a spontaneous osteochondroma developed in a female Sprague–Dawley rat that, after repeated in vivo passages, acquired CS features. Different cell lines have been derived from this model such as SRC-JWS, endowed with aggressive behavior, high tumorigenicity, and fast growth, or the slow-growing SRC-TRO cell line. SRC tumor tissues have been used for subcutaneous or orthotopic allotransplantation in Lewis rats or Wistar rats with some difference in growth related to their different levels of thyroid hormone [72,73]. A valuable advantage of the SRC model is the possibility to evaluate SRC growth and metastasization in relation to tumor microenvironment in an immunocompetent host. In SRC, the immune microenvironment affected CS tumor growth, depletion of T lymphocytes resulted in increased tumor growth rates, while depletion of CD163^+^ macrophages delayed tumor progression [74].

Limitations of the SCR model could be ascribed to genetic alterations that differ from those observed in human CS, reducing its relevance and applicability to the study of human pathology.

Other transplantable animal CS models were derived from spontaneously arisen hamster CS models (Syrian hamster and Chinese hamster) [72].

### 4.6. CS Transgenic Mouse Models

Expression of activated oncogenes in transgenic mice has been shown to be able to induce the onset of several tumors.

CS development was reported in c-fos transgenic mice. The expression of this oncogene elicited bone neoplastic lesions that in half of the cases, after a long latency time (up to 9 months), evolved to CS. Tumor induction was dependent on a replacement of 3′ noncoding sequences of c-fos by a long terminal repeat (LTR) of the murine retrovirus FBJ-MSV [75,76]. The peculiar genetic base, long latency time, and incomplete penetrance of CS onset limited the utility of the model for CS studies.

The development of enchondromas was induced in transgenic mice overexpressing the Gli2 transcription factor. When these mice were crossed with p53 deficient mice, the loss of one p53 allele gave rise to low-grade CS in 50% of the mice, or high-grade CS showing loss of both p53 alleles in 6% of the mice [77].

By introducing in mice the IDH1 R132Q mutated gene, Hirata et al. [31] were able to induce enchondromas in mice. By using the Cre–Lox technology [78] for conditional gene expression, they found that Col2a1-Cre;Idh1-KI mice did not survive after birth due to tracheal cartilage defects and diffused cartilage alterations. Hirata et al. then developed a viable tamoxifen-inducible Col2a1-Cre/ERT2;Idh1-KI transgenic mouse line in which tamoxifen was administered at four weeks of age. At 3–6 months after tamoxifen treatment multiple enchondroma-like cartilage lesions were observed, but no progression to CS occurred, thus highlighting the need for additional mutations for CS onset. Conversely, the development of CS was reported in the conditional mouse model Col2-Cre;Trp53^f/f^/Rb1^f/f^. When double conditional deficiency of Trp53 and Rb1 driven by a Col2a1 promoter was induced in chondrocytes, all the animals showed spinal CS onset starting from 1 month of age. Lung metastases were also observed, and mice survival did not exceed 1 year. Compared to chondrocytes carrying each single deletion of Trp53 or Rb1, double Trp53/Rb1-deficient chondrocytes showed a significantly increased expression and transcriptional activity of Yes-associated protein (YAP), a transcriptional coactivator. When the triple conditional knockout mouse model Col2-Cre;Trp53 f/f/Rb1f/f/YAP f/f was obtained, the additional silencing of YAP could hamper CS progression and lung metastasis. Furthermore, metformin, which is a YAP inhibitor, was able to significantly reduce CS formation and progression in vivo in the double Trp53/Rb1 knockout model, thus indicating YAP inhibition as a potential therapeutic strategy for CS [22].

Peripheral CS carry EXT1–2 mutations and frequently additional disruptions of *CDKN2A* or *TP53*. When the conditional loss of *Trp53* or *Ink4a/Arf* was added in an *Ext1*-driven mouse model of osteochondromagenesis, the silencing of each single tumor suppressor gene induced the development of peripheral CS [6]. Overall, transgenic CS modeling provided proof of concept for the role of specific genetic alterations in CS pathogenesis, but late CS onset and incomplete penetrance limited their general utility in preclinical drug screening research, an aim that was better achieved by human-derived CS immunodeficient mouse models.

### 4.7. Human-Derived CS Xenograft Models in Mice

Immunodeficient mice showing various degrees of immunodeficiency, athymic nude mice, severe combined immunodeficient (SCID), non-obese diabetic SCID gamma (NSG) mice [79], have enabled human xenograft transplantation. Immunodeficient mice accept also engraftment of tumor fragments directly derived from patients and are defined as patient-derived xenografts (PDXs). High tumor take is mainly observed in high-grade CSs, or p53 mutated, or dedifferentiated tumors, while low-grade CS shows no tumor take or very slow in vivo growth. Studies with low-grade tumors can require long-term experiments and this may discourage their employment in drug screening experiments with a lower representation of the clinical heterogeneity and a lack of information on drug activity or resistance in slow-growing tumors.

### 4.8. CS Cell Line-Derived Xenograft Models in Mice

Several human CS cell lines have been successfully used to generate tumors in immunodeficient mice after subcutaneous or orthotopic injection, as reported above in Table 2. One of the most used is the human CS cell line JJ012, which was used to establish different subcutaneous and orthotopic xenograft models in BALB/c nude mice. The periosteal implant in the tibia gave rise to lung micrometastases in 100% of the mice while the intramedullary implant produced lung micrometastases in half of the mice [36,52].

### 4.9. CS Patient-Derived-Xenograft (PDX) Models

Bone sarcoma PDX collections reported in the literature usually include few CS PDX models and it is difficult to estimate the rate of successful engraftment, which seems to be very low. In the study of Monderer et al. [60], only 1 out of 10 (10%) CS biopsy implanted subcutaneously in nude mice gave rise to a PDX that could be propagated in vivo maintaining the phenotype of the original patient (a conventional CS grade 2). In a different study, aimed at defining the metabolic profile of IDH mutant and non-mutant CS, a large panel of 17 CS PDX in NSG mice was obtained. The use of PDX tumor samples allowed the analysis of CS tumor metabolism eliminating confounding factors of in vitro cell culture or patient habits variability. The panel was representative of the clinical situation and included IDH wild-type and IDH1/2 mutant CS. Tumor histotypes collected were eight central CS, seven dedifferentiated CS, and two clear cell CS. Mutant IDH CSs showed a peculiar metabolic profile compared to wild-type. The signature of IDH mutant CSs was a global increase of amino acids, lactate accumulation, and elevation of acylcarnitines. Unfortunately, data on CS PDX rate of engraftment, growth characteristics, or stabilization over two or three in vivo passages, cryopreservation, and availability were not provided [80].

To the best of our knowledge, very few clinically annotated CS PDX models are reported across several studies and public or commercial PDX repositories. One CS PDX (CF8X) model is present in the patient-derived model repository of the National Cancer Institute (NCI-PDMR) (https://pdmr.cancer.gov) and one (WU-0064) in the PDXNet Consortium, both were used for extensive genomic and molecular studies [81]. Two conventional CS PDX models, CTG-1255 and CTG-2383 are commercially available at Champions Oncology, Inc. (Hackensack, NJ, USA) [82], and eight conventional CS PDXs are available at the Jackson Laboratory for drug screening studies [83]. The Xenosarc collection includes a mesenchymal CS [84], and another model of mesenchymal CS carrying the HEY1::NCOA2 gene fusion was established by Safaric Tepes and colleagues [83]. Two conventional CS PDX models, CS-347 and CS-281, grade 2 and grade 3, respectively, have been established in the study of Giordano et al. [85], which showed in both CS PDXs and in the CS-281 PDX-derived cell line a high expression of Ephrin type A receptor 2 (EphA2) and an in vitro dose-dependent growth inhibition by the EphA2 inhibitor ALW II-41-27, suggesting a new possible targeted therapy for CS.

A comprehensive overview of CS preclinical models is reported in Figure 1. Altogether, CS models are still scarce and are far from completely capturing the high CS heterogeneity in clinics. Even if dealing with slow-growing tumors and long-lasting experiments, further efforts should be made to this end.

## 5. Innovative Therapies Currently Under Investigation in CS

Currently, no US Food and Drug Administration (FDA)-approved therapies exist for conventional CS, and no uniform treatment lines are accepted for advanced CS patients. Advantages deriving from adjuvant chemotherapy in CS are still controversial and not fully established [14]. This generates an unmet need for new and effective treatment options in case of refractory, inoperable, and metastatic CS. The search for new therapies is moving in several directions and is exploring several opportunities, ranging from targeted therapies, epigenetic therapies, immunotherapies, or combinations, including therapies aimed at increasing chemotherapeutic drug sensitivity. In this section, we summarize recent advancements in CS therapeutic approaches.

### 5.1. Mutant IDH Targeted Therapies

The inhibition of mutant IDHs in various tumors has been considered of great interest and, based on clinical trial results, FDA approval was achieved for selected IDH-mutated tumors [28,86]. Enasidenib, an IDH2 inhibitor, and ivosidenib (or AG-120), an IDH1 inhibitor, were approved by the FDA in 2017 and 2018, respectively, with an indication for IDH-mutated AML [29,87]. In 2021, ivosidenib received approval for IDH1-mutated cholangiocarcinoma [88], and in 2024, vorasidenib, a dual inhibitor of mutated IDH1/2, received FDA approval for glioma [89].

Clinical evaluation of ivosidenib in CS is ongoing. A phase I trial of ivosidenib in 21 IDH-mutated CS patients (ClinicalTrials.gov ID: NCT04278781) resulted in a median progression-free survival (PFS) of 5.6 months, where the best overall response was stable disease in 52% of the patients, and no complete responses or partial responses were observed during the first two years [90]. As a follow-up to the original study, patients with disease stability beyond 2 years were found to have an overall response rate of 23.1%. Of note, prolonged progression-free survival could be achieved in a subset of patients with advanced CS, mainly in patients with a minimal number of co-occurring mutations. Moreover, PFS at 3 and 6 months was 30% and 0% for dedifferentiated CS and 77% and 54% for conventional CS, thus suggesting a different sensitivity between the two CS subtypes despite a similar IDH mutational status [27]. A phase II clinical trial (ClinicalTrials.gov ID: NCT04278781) is ongoing to evaluate ivosidenib in locally advanced, metastatic, or recurrent grade 2 or grade 3 IDH1mut CS. A phase III clinical trial, the CHONQUER study, was registered in 2023 (ClinicalTrials.gov ID: NCT06127407; Study CL3-95031-007) and it is now recruiting patients for evaluation of orally administered ivosidenib in patients with IDH1-mutated, locally advanced or metastatic conventional CS grades 1, 2, or 3 and not eligible for curative resection. The conclusion of the study is scheduled for 2028–2030 (NCT06127407) (accessed on 11 November 2024).

Finally, the evidence that gliomas harboring IDH1 R132H mutation carry an immunogenic epitope that can induce a CD4+ T-helper/T-cell response able to hamper tumor growth through the production of pro-inflammatory cytokines such as interferon-γ and TNF-α, lead to the development of vaccines against IDH1 mutant epitopes. Some of these vaccines are being investigated in phase I clinical trials in gliomas, for example, the NOA21 trial (ClinicalTrials.gov ID: NCT03893903) aims at assessing the safety and immunogenicity of IDH1-vac in combination with the programmed death-ligand 1 (PD-L1)-blocking immune checkpoint inhibitor avelumab in gliomas [29]. Results of the vaccine trials will be of great interest also for other IDH1 mutated tumors, although in CS, at difference with glioma, the prevalent IDH1 mutation is R132C.

### 5.2. HIF-2a and SIRT1 Targeted Therapies to Enhance CS Drug Sensitivity

HIF-2α is frequently overexpressed in high-grade CS and is associated with poor prognosis. Targeting of HIF-2α with the small inhibitor TC-S7009 inhibited tumor growth in two different CS xenograft models obtained after orthotopic or subcutaneous injection of the SW1353 and JJ012 CS cell lines, respectively. After combination with cytotoxic drugs such as cisplatin or doxorubicin, the antitumor response was strongly enhanced highlighting a synergistic effect [36].

HIF-2α inhibition sensitized CS cells to apoptosis induced by DNA-damaging drugs and could represent a strategy to restore drug sensitivity in CS cells. Two TC-S7009 analog HIF-2α antagonists have been proposed for clinical use in clear cell renal carcinoma showing increased HIF-2α levels. The study of Kim and coworkers [36] provided a rationale for implementing HIF-2α inhibition strategies in the treatment of CS patients.

As described above, cancer cells frequently show dysregulated NAD^+^ metabolism, but its targeting represents an open challenge due to the broad involvement of NAD^+^ in normal biological processes. Cancer cells can develop strong NAD^+^ dependencies but are able to upregulate and exploit different NAD^+^ synthetic pathways. CSs with a dismal prognosis exhibit increased NAD^+^ biosynthesis that is regulated by the SIRT1-HIF-2α axis. Inhibiting SIRT1 exposes CS cells to vulnerabilities imposed by NAD^+^ dependencies and is critical for CS cell survival, inducing a synergistic cytotoxic effect when combined with chemotherapy. The SIRT1 inhibitor EX527 in combination with doxorubicin exerted a potent antitumor activity in multiple CS xenograft models. The good results obtained in the preclinical setting warrant further investigation in CS near-patient models and possibly clinical development [38].

### 5.3. Death Receptor 5 (DR5) Targeting in CS

Triggering of death receptor 5 (DR5), a proapoptotic multimeric cell surface receptor that is activated after binding tumor necrosis factor-related apoptosis-inducing ligand (TRAIL), is a potent inducer of apoptosis in tumor cells but not in normal cells [91], and remarkable antitumor activity has been demonstrated also in CS. A study using the recombinant agonist Apo2L/TRAIL (dulanermin) for the dual proapoptotic receptor DR4/DR5 showed strong and long-lasting antitumor activity in a metastatic CS patient [92], but rapid clearance and decoy receptor binding limited its efficacy. Further affords have been done to target DR5, such as the multimeric anti-DR5 IgM agonist antibody IGM-8444, which is now in phase I clinical trial for hematological and solid tumors including CS (NCT04553692) [93]. Recently a third-generation, recombinant, humanized, agonistic antibody against DR5, named INBRX-109, has been designed to achieve selective DR5 agonism and safety. INBRX-109 is a tetravalent single-domain antibody that engages four DR5 receptors and presents the best rate of activity against tumor cells compared to normal cells. A significant single-agent antitumor activity in two conventional CS PDX models (CTG-1255 and CTG-2383) paved the way to the first-in-human phase I study in patients with advanced solid tumors (NCT03715933) including CS. In the single-agent expansion part of the study, INBRX-109 showed activity regardless of the IDH1/IDH2 mutational status. The disease control rate (stable disease and partial response) was 87.1%, with a median PFS of 7.6 months, similar to that achieved with pazopanib (7.9 months), but increased compared to those observed with other treatments, where PFS was 4, 5, or 5.6 months with chemotherapy, regorafenib, or ivosidenib, respectively [82]. Durable clinical benefit was observed in 40.7% (11 of 27) patients, including two PRs [94]. A randomized, placebo-controlled, phase II trial, the ChonDRAgon trial (NCT04950075), will further evaluate INBRX-109 efficacy in unresectable or metastatic conventional CS and is now recruiting patients in the USA and European countries. Of note, in the phase I clinical trial described above, DR5 expression in CS tumor tissue was not assessed. Since the immune infiltrate in CS is mainly composed of immunosuppressive macrophages that express DR5, a possible mechanism of action of INBRX-109 could be related to the impairment of the suppressive immune cell population within the tumors or a dual targeting of tumor cells and immunosuppressive cells, but further investigation on mechanisms of action is required.

Many other targeted therapies have been evaluated in CS, including conventional tyrosine kinase inhibitors imatinib and regorafenib, the cyclin-dependent kinase inhibitor abemaciclib, and the hedgehog inhibitor saridegib. Although these drugs are currently approved for other cancers, they did not show sufficient activity in CS [82].

### 5.4. Antiangiogenic Therapies

Vascular endothelial growth factor A (VEGF-A) is a key inducer of angiogenesis in many solid tumors. VEGF-A, produced by tumor cells, binds to the transmembrane tyrosine kinase receptors VEGFR-1 and VEGFR-2 on endothelial cells and increases their survival, migration, and proliferation. VEGF-A overexpression has also been observed in CS with higher expression in grade 2 as compared to grade 1 [62]. Pazopanib is a multiple tyrosine kinase inhibitor targeting VEGF receptors, platelet-derived growth factor (PDGF) receptors, and KIT. A phase 2 clinical trial (NCT01330966) investigated its efficacy in unresectable and metastatic conventional CS and reported a positive activity with a disease control rate (SD and PR only) of 43%, a median PFS of 7.9 months and a median overall survival of 17.6 months [95]. Similarly, regorafenib, a VEGF receptor targeting drug, was able to prolong progression-free survival of advanced CS patients [96] suggesting a possible role for pazopanib and other VEGFR-targeting drugs in the therapy of CS. Clinical trials combining antiangiogenic drugs, immune checkpoint inhibitors, and chemotherapy have been proposed [97].

### 5.5. Chondroitin Sulfate Proteoglycan 4 Immunological Targeting

Chondroitin sulfate proteoglycan 4 (CSPG4) is a proteoglycan that is expressed at high levels on the cell surface of several tumor types and sarcomas including CS. Its limited expression in normal tissues makes it an appropriate target for immunological targeting. In a cohort of 76 patients with CS, 71% of conventional CS samples showed high-medium expression of CSPG4 with the largest percentage of positive samples in grade 2, while only 15% of dedifferentiated CSs displayed CSPG4 expression. Medium and medium-high CSPG4 expression was associated with a shorter time to metastasis and reduced overall survival in conventional and dedifferentiated CS, respectively [98]. Genetically engineered CSPG4-targeted chimeric antigen receptor T (CAR-T) cells were effective in killing CS cells in vitro, suggesting that CSPG4-targeted CAR-T cells could represent an effective strategy in the treatment of CSPG-4 expressing CS [98]. This approach deserves further investigation taking into account that even active immunization strategies against CSPG4 are in development and are effective in deferring tumor growth and metastasis in osteosarcoma preclinical models [99].

### 5.6. Immunotherapeutic Approaches in CS

The interaction of the immune checkpoint programmed cell death 1 (PD-1) receptor expressed on T cells with its ligand PD-L1, expressed on tumor cells, macrophages, endothelial cells, and other immunosuppressive cells, elicits an inhibitory signal in T cells, inducing anergy and suppression of immune surveillance. Immunohistochemical studies revealed that PD-L1 was expressed by approximately 50% of dedifferentiated CS, but was absent in conventional, mesenchymal, and clear cell CS [100]. Expression of PD-L1 in dedifferentiated CS, but not in other CS subtypes, was confirmed in other studies, and overall survival or metastatic condition of patients was not correlated with PD-L1 expression [101]. Dedifferentiated CS tumors expressing PD-L1 were highly infiltrated by T cells, but no differences in T-cell infiltrate or PD-L1 expression were detected between IDH mutant and IDH wild-type tumors. Most dedifferentiated CS (85%) were also infiltrated by immunosuppressive macrophages and downregulation of HLA class I expression was observed in 55% of the dedifferentiated CS [100]. The presence of an immune infiltrate and of PD-L1 expression provided a rationale for the use of immune checkpoint inhibitors in CS patients. However, clinical trials with checkpoint inhibitors in sarcoma patients produced controversial results, reporting an overall limited efficacy but suggesting, as regards CS, the need for further evaluation in larger cohorts of patients. The phase II clinical trial SARC028 (NCT 02301039), investigating anti-PD-1 antibody pembrolizumab in soft tissue and bone sarcoma, found that, out of five advanced CS patients, one had an objective response, one had stable disease, and three patients had progressive disease [102]. In a study assessing the combination of doxorubicin and pembrolizumab in sarcomas and enrolling five CS patients, three had partial responses [103]. In addition, several reports of sporadic responses to checkpoint inhibitors [104,105] suggested the need for studies with larger cohorts of CS patients and strategies for better stratifying patients.

Several recent studies found a correlation between immune infiltrate composition, tumor aggressiveness, response to immunotherapy, and survival in CS. A study employing fresh tissue multi-omic profiling addressed the potential of a classification based on the immune infiltrate in predicting the outcome of conventional CS in response to immunotherapy [47]. In a cohort of 20 patients, three distinct immune subtypes were identified: subtype I, rich in granulocytic myeloid-derived suppressor cells (G_MDSC) inducing an immunosuppressive environment, named “G-MDSC dominant”; subtype II, showing an infiltrate dominated by exhausted T cells and dendritic cells, indicated as “immune exhausted”; and subtype III, poorly infiltrated by immune cells and defined as “immune desert”. Of note, this study reported a very low presence of macrophages, monocytes, NK cells, and B cells in CS infiltrate, and the majority (60%) of the CS samples belonged to the subtype “immune desert”. Previous studies found no difference in tumor infiltrate between IDH-mutated or wild-type tumors [100], but in this study, IDH1/2-mutated CSs were assigned to the immune cell–rich subtypes (subtype I and II). IDH1/2-mutated CSs had significantly more immune cells and a more inflamed microenvironment, with higher levels of CXCL9 and CXCL12, as compared to IDH1/2 wild-type CS. IDH mutations elicited a favorable microenvironment for a potential immune response by increasing the levels of chemokines mediating immune cell homing. In a small cohort of 12 CS patients treated with checkpoint inhibitors (anti-PD-1 antibodies such as pembrolizumab or sintilimab), the only 3 patients that had partial response or stable disease belonged to the “immune exhausted” subtype [47].

In other studies, macrophages were reported as the main component of the leukocyte infiltrate, up to 75% of infiltrated immune cells [43,101]. Tumor-infiltrating lymphocytes (TIL) and tumor-associated macrophages (TAM) were positive and negative prognostic factors, respectively, in dedifferentiated CS. The high density of CD3+ and CD8+ lymphocytes correlated with longer overall survival, while a high ratio of CD68+ macrophages/CD8+ lymphocytes and higher concentrations of macrophage chemoattractants in tumors were shown to correlate with poor prognosis. TAM distribution was peritumoral in conventional CS and more abundant in dedifferentiated areas in the dedifferentiated subtype [101]. Metastatic patients at diagnosis presented a higher density of CD68+ and CD163+ M2 polarized, pro-tumoral immunosuppressive TAM compared to patients with a localized disease. Of note, colony-stimulating factor 1 receptor (CSF1R), also known as macrophage colony-stimulating factor receptor (M-CSFR), promoting macrophage recruitment and proliferation, was expressed by TAM in about 90% of dedifferentiated CS and 60% of conventional CS. The high presence of macrophages in different CS subtypes argues in favor of macrophages-targeting therapies to achieve therapeutic efficacy in CS, such as the use of CSF1R inhibitors [101,106] or antibodies depleting specifically M2 macrophages or myeloid suppressive cells by targeting the triggering receptor expressed on macrophages 2 (TREM2) [107]. Altogether, these data suggest the need for further studies to establish the ability of an immune-based classification to predict the response to immunotherapy and the utility of checkpoint inhibitors in the treatment of CS.

### 5.7. Epigenetic Therapies

The increase in DNA and histone methylation induced by IDH1/2 mutations, although with lower hypermethylation and different hypermethylated sites in IDH-mutated dedifferentiated CS compared to IDH-mutated conventional CS [27], provided a rationale for investigating the role of therapies based on epigenetic modulation in these CS subtypes. Epigenetic drugs include both DNA hypomethylating agents, such as DNA methyltransferase (DNMT) inhibitors, 5-aza-2′-deoxycytidine (5-aza) and decitabine, and histone deacetylase (HDAC) inhibitors, for example, vorinostat, romidepsin, belinostat, and panobinostat, which have received FDA regulatory approval for hematological malignancies. Preclinical evaluation of a combination of 5-aza and vorinostat in CS showed potent antitumor activity in vitro and in vivo in the JJ012 cell line-derived xenograft model. Notably, the combination treatment was significantly more effective than either monotherapy. Elevated DNA damage response, expression of interferon-stimulated genes, including PD-L1, and innate immune response were observed. Based on these data, a phase II clinical trial evaluating the DNMT inhibitor guadecitabine (SGI-110) in combination with the HDAC inhibitor belinostat in patients with unresectable or metastatic conventional CS (clinicaltrial.gov ID: NCT04340843) was undertaken but failed to meet the primary endpoint of overall response rate (ORR) [108]. The next step will further evaluate the combination of epigenetic therapy with immune checkpoint inhibitors and chemotherapy [109,110]. The main innovative therapies under evaluation in clinical trials are listed in Table 3.

## 6. Conclusions

Addressing the intrinsic resistance of chondrosarcoma (CS) to chemotherapy and radiotherapy, particularly in advanced, unresectable, or metastatic cases remains a critical and unresolved clinical challenge. Recent advancements in multi-omic approaches highlighted the complex pathogenesis of CS, stressing the pivotal role of the tumor microenvironment in patient outcomes and response to immunotherapy. Actionable mutations, such as IDH1/2 alterations, further underscore the need for robust stratification strategies to optimize personalized treatments.

The development of multi-omic signatures, based on mutational status and immune microenvironment profiles, offers the potential to refine patient stratification, enabling tailored therapeutic regimens that improve treatment efficacy and survival outcomes. The integration of these signatures into clinical practice could redefine the standard of care, offering novel therapeutic options for CS patients.

Preclinical modeling, particularly the use of patient-derived xenografts (PDXs) and PDX-derived cell lines, is a vital platform for advancing drug discovery. These models faithfully replicate the genetic alterations of primary tumors, especially in high-risk and treatment-resistant CS subtypes, providing a valuable resource for drug development and prioritization. To maximize their utility, efforts should focus on establishing large, shared repositories of well-characterized PDXs and annotated genomic datasets. This infrastructure will facilitate genome-matched therapeutic decisions and support multi-omic-guided interventions.

The extreme genetic heterogeneity and rarity of CS, coupled with the unsatisfactory results of previous clinical trials, emphasize the urgency for next-generation clinical trials which should integrate genomic data and functional studies to capture the dynamic evolution of individual tumors. By coupling advanced preclinical models with personalized multi-omic approaches, the field can move closer to overcoming CS’s therapeutic challenges and improving outcomes for aggressive CS subtypes.

## Figures and Tables

**Figure 1 ijms-26-01542-f001:**
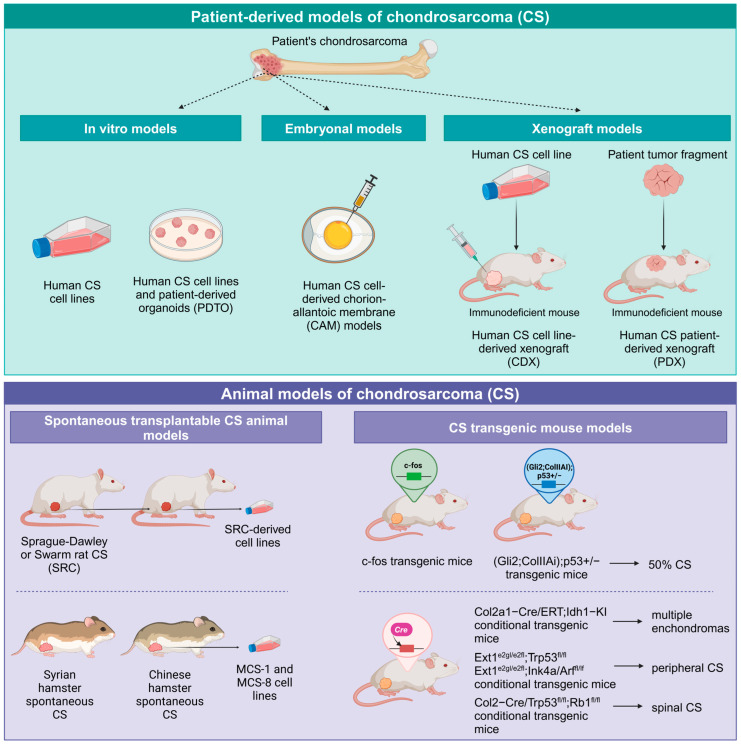
CS preclinical models. Experimental models that have been generated for CS can be distinguished into patient-derived models and animal models. Patient-derived models (upper panel) include cell lines, organoids and 3D cultures, embryonal models established using chorion-allantoic membranes, and xenograft models, both CDX and PDX, established using immunodeficient mice. Animal models (lower panel) range from spontaneous transplantable CS models of rats and hamsters to genetically modified transgenic or conditional mouse models carrying different combinations of genes involved in the pathogenesis of CS (Created in BioRender. K.S. (2025) https://BioRender.com/n24l231, accessed on 8 February 2025).

**Table 1 ijms-26-01542-t001:** CS subtypes and main clinical features.

CS Type	Origin (% of CS Cases)	Location in Bone	Main Genetic Alterations	5-Year Survival Rate
Conventional CS [3,4,5,14]	Bone, Primary (85%)	Central (great majority)	>50% IDH1/2 mutation	ACT: 93% Grade 1: 90% Grade 2: 80% Grade 3: 29%
		Periosteal (Juxtacortical) (1%)	IDH1/2 mutation Altered hedgehog pathway	68–93%
	Bone, Secondary (<10%)	Central (deriving from enchondromas)	>50% IDH1/2 mutation	>90%
		Peripheral (deriving from osteochondromas)	EXT1/2 mutation	>90%
Dedifferentiated CS [7,14]	Bone (10–15%)		80% IDH1/2 mutation	10–25%
Clear cell CS [7,9,10,14]	Bone (2–6%)		No specific driving mutations	62–100%
Mesenchymal CS [7,8,14]	Bone or soft tissues (2%)		HEY1::NCOA2 or IRF2BP2::CDX1 translocations	37–50%
Extraskeletal myxoid CS [11,12]	Soft tissues (3% of soft tissue sarcomas)		EWS::NR4A3, TAF15::NR4A3, or TCF12::NR4A3 translocations	50%

**Table 2 ijms-26-01542-t002:** Most used human CS cell lines and their biological characteristics.

CS Cell Line	Parental Tumor (Gender)	Main Genetic Features	Tumorigenicity	Ref.
JJ012	Grade 2 conventional CS (male)	IDH1(R132G) heterozygous mutation	Yes, nude mice; subcutaneous (sc), and orthotopic tumors	[52]
L835	Grade 3 central conventional CS (male)	IDH1(R132C) heterozygous mutation; homozygous deletion of the CDKN2A	No	[50]
SW1353	Grade 2 conventional central CS (female)	IDH2(R172S) heterozygous mutation	Yes, SCID and nude mice; sc and orthotopic tumors; experimental metastasis after intrasplenic injection	[37]
OUMS27	Grade 3 central conventional CS (male)	IDH1/2 wild-type, p53 mutated	Yes, nude mice	[46]
HCS-2/A and HCS-2/8	Grade 1 Well-differentiated CS (male)	Not reported	HCS-2/A highly tumorigenic in nude mice; HCS-2/8 low tumorigenicity	[45]
HCS-TG	Grade 2 CS (male)	Not reported	Yes, nude mice	[53]
NDCS-1	Dedifferentiated CS, metastasis (male)	IDH1/2 wild-type, p53 mutated	Yes, SCID mice	[54]
L2975	Dedifferentiated CS, metastasis (male)	IDH2R172W heterozygous mutation; homozygous deletion of the CDKN2A	Yes, nude mice, slow-growing tumors	[50]
L3252	Dedifferentiated CS, local recurrence (female)	IDH1/2 wild-type; homozygous deletion of CDKN2A	No	[50]
CH-3573	Grade 2 conventional central CS (male)	IDH1/2 wild-type	Yes, nude mice	[55]
CH-2879	Grade 3 conventional central CS	IDH1/2 wild-type	Yes, nude mice	[49,56]
CDS17 and T-CDS17	Dedifferentiated CS (male)	IDH2R172G mutation	Yes, NOD/SCID mice, slow-growing tumors	[57]
SMU-DDCS	Dedifferentiated CS (female)	IDH1 mutation	Yes, nude mice	[58]
CHSA or CS1	High grade CS (male)	IDH2 mutation	Yes, nude mice Highly tumorigenic	[59]
CH03	Dedifferentiated CS (female)	IDH1/2 wild-type; TP53 deletion; p16^ink4a^ deletion	No, nude mice	[60]
CH34	Conventional CS grade 3 (male)	IDH1 mutation; TP53 wild-type; p16^ink4a^ deletion; p14^ARF^ deletion; MDM2 amplification	No, nude mice	[60]
CH56	Conventional CS grade 3 (female)	IDH2 mutation; TP53 wild-type; p16^ink4a^ deletion; p14^ARF^ deletion; MDM2 amplification	Yes, nude mice	[60]
CAL78	Dedifferentiated CS (male)	IDH1/2 wild-type	Yes, nude mice	[61]
C3842	Secondary CS (male)	IDH1/2 wild-type	Not reported	[62]

**Table 3 ijms-26-01542-t003:** Main innovative therapies for CS and published clinical trial results.

Therapeutic Approach	Clinical Trial *	Drugs	Main Clinical Results	Ref.
Inhibition of mutant IDH1/2	NCT04278781 Phase I/II trial IDH1 mutated advanced CS patients	Ivosidenib (IDH1 inhibitor)	Median PFS of 5.6 months. Best overall response: SD in 52% of the patients, PFS at 6 months was 0% for dedifferentiated CS and 54% for conventional CS	[90]
	NCT06127407 CHONQUER study Phase III trial		Results awaited in 2028–2030	[90]
DR5 targeting	NCT04553692 Phase I clinical trial	IGM-8444 (multimeric anti-DR5 IgM agonist antibody)	Estimated study completion by the end of 2027	[93]
	NCT03715933 Phase I trial	INBRX-109 (tetravalent single-domain agonistic antibody against DR5)	Disease control rate (SD and PR) was 87.1% with a median PFS of 7.6 months; durable clinical benefit in 40.7% (11 of 27) patients, including two PRs	[82]
	ChonDRAgon trial NCT04950075 Phase II trial		Estimated study completion by the end of 2026	[82]
Antiangiogenic therapies	NCT01330966 Phase II trial	Pazopanib	Disease control rate (SD and PR) 43%, median PFS of 7.9 months, PFS rate at 6 months 55%, median overall survival 17.6 months	[95]
	NCT02389244 Phase II trial	Regorafenib	Median PFS of 5 months, PFS rate at 6 months 43%	[96]
	NCT04055220 Efficacy and safety of Regorafenib as maintenance therapy		Recruiting, estimated study completion by the end of 2026	[96]
Immunotherapy and combined therapies	SARC028 NCT 02301039 Phase II trial	Pembrolizumab anti-PD-1 antibody	1PR and 1 SD out of 5 CS patients	[102]
	IMMUNOSARC study NCT03277924 Phase II trial	Nivolumab and sunitinib (anti-PD-1 antibody + antiangiogenic drug)	In the dedifferentiated CS cohort median PFS of 5.6 months, PR 26.3%, and 52.6% SD	[94,110]
Epigenetic therapies	NCT04340843 NCI 10330	Belinostat with SGI-110 or ASTX727 (HDACinhibitor + DNMTinhibitor)	Failed to meet the endpoint of ORR	[108]

* NCT numbers from http://clinicaltrials.gov (accessed on 31 January 2025). PFS, progression-free survival; SD, stable disease; PR, partial response; ORR, overall response rate.

## Data Availability

Not applicable.

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
