# Peer review of "Chondrosarcoma: New Molecular Insights, Challenges in Near-Patient Preclinical Modeling, and Therapeutic Approaches"

_ijms, 2025, doi:10.3390/ijms26041542_

Round 1
Reviewer 1 Report
Comments and Suggestions for Authors
Summary:
This manuscript reported a review on the chondrosarcoma (CS) from many aspect including new molecular studies, near-patient preclinical modeling, as well as potential therapeutic approaches. As mentioned in the text, chondrosarcoma family includes various tumor entities with various peculiar biological, genetic, and epigenetic characteristics and clinical behaviors. The spectra of pathology subtypes include conventional CS, high-grade, dedifferentiated and mesenchymal CS, as well as unresectable and metastatic CS. Novel therapeutic strategies are urgently needed because of its high resistance to chemotherapy and radiation therapy. This review is comprehensive and address well the current landscape of in vitro and in vivo CS models, focusing on the recent development of new targeted therapies against CS, from aspects including IDH1/2 mutations, NAD+ dependency and SIRT1-26 HIF-2α axis, or exploring DR5 targeting and immunological approaches. However, some issues needed to be addressed before considering to be published in IJMS.
1. Local recurrence is common in patients with chondrosarcoma, especially for those with high grade malignancy and in location hindering from radical resection. Can the authors address this issue?
2. Lung metastasis is a main cause of mortality for the patients with chondrosarcoma. More discussion of this issue should be included in this review. The authors may present the strategies to control or decrease the impact of lung metastasis.
3. Table 1, the authors should listed the references.
4. Table 2, Could authors recommend the molecular pathological diagnosis strategy for differentiating the subtypes of chondrosarcoma in the clinical practice?
5. Regarding the innovative therapies currently under investigation in CS, authors may need to list the preliminary clinical outcomes.
Reviewer 2 Report
Comments and Suggestions for Authors
This review provides a detailed discussion of the clinical and pathological classification of chondrosarcoma, commonly used cellular and animal experimental models, as well as advancements in clinical treatment. It summarizes recent progress in the basic and clinical research of this disease and its therapeutic outcomes. The following issues are identified and suggested for revision:
1. The content is overly lengthy, particularly in the sections on CS preclinical models and near-patient models. It is recommended to condense these parts.
2. The abstract requires revision as it fails to accurately summarize the content of the article.
3. References are missing in the "CS subtypes and main clinical features" .
Round 2
Reviewer 2 Report
Comments and Suggestions for Authors
I believe the author has made sufficient revisions and content additions to the article, and I recommend it for publication.